# The Effects of Aerobic Exercise on Executive Function: A Comparative Study Among Active, Passive, and Non-Procrastinating College Students

**DOI:** 10.3390/bs15020225

**Published:** 2025-02-17

**Authors:** Chentao Liu, Juanjuan Zhang

**Affiliations:** 1Department of Physical Education, Northwest University, Xi’an 710127, China; 2Department of Physical Education, Xi’an International Studies University, Xi’an 710128, China; zhangjuanjuan@xisu.edu.cn

**Keywords:** aerobic exercise, active procrastination, passive procrastination, non-procrastination, executive function

## Abstract

Objective: This study aims to explore the effects of an aerobic exercise intervention on the executive functions of active, passive, and non-procrastinating college students. Methods: A total of 190 college students (36 male, 154 female, 19.56 ± 1.11 years old) with different types of procrastination were recruited from the first and second years of a university using the General Procrastination Scale and the Active Procrastination Scale. A 3 (procrastination type: active procrastination, passive procrastination, non-procrastination) × 2 (group: exercise group, control group) × 2 (measurement time: pre-test, post-test) mixed experimental design was employed. All participants attended regular physical education classes as usual, while the exercise group participated in an 8 week aerobic exercise program. Before and after the intervention, the inhibition, updating, and switching sub-functions of executive function were assessed. Results: (1) Active procrastinators and passive procrastinators showed significant differences in their inhibition scores, inhibition accuracy, updating scores, and updating accuracy (*p* < 0.05). Non-procrastinators exhibited a significantly higher inhibition accuracy than passive procrastinators (*p* < 0.05), while active procrastinators had a significantly higher updating accuracy than non-procrastinators (*p* < 0.05). As can be seen, there were significant differences in inhibition and updating functions between active procrastinators and passive procrastinators, with passive procrastinators showing obvious deficiencies in their inhibition accuracy. (2) After the intervention, the aerobic exercise group of non-procrastinators showed a significant reduction in their inhibition reaction time and updating reaction time (*p* < 0.05). The passive procrastination in the exercise group showed significant improvements in switching accuracy and inhibition accuracy (*p* < 0.05). The active procrastination in the exercise group showed a significant reduction in updating reaction time (*p* < 0.05). From the above findings, it is clear that the eight-week aerobic exercise intervention has shown improvement effects on the executive function of college students with different procrastination types, and the extent of the improvement in each sub-function of executive function varies depending on the type of procrastination. Conclusion: This study reveals the relationship between exercise and the development of the sub-functions of executive function in college students with procrastination, further validating the effectiveness and feasibility of implementing exercise interventions in real-world school settings.

## 1. Introduction

With the rapid development of modern society, the issue of procrastination has become more severe and prominent. Procrastination is often regarded as a manifestation of self-regulation failure ([41]). Surveys indicate that 30–60% of college students frequently procrastinate ([30]). Based on the effects of the behavior, procrastination can be categorized into active procrastination and passive procrastination. Passive procrastinators often fall into a vicious cycle due to a lack of effective self-regulation, resulting in poor task completion quality and higher levels of anxiety and stress ([5]). Although both active and passive procrastinators exhibit irrational delays, active procrastinators demonstrate a greater focus in the later stages of tasks and are less affected by negative emotions compared to passive procrastinators ([4]), leading to better academic performance ([36]). Therefore, in-depth research into the cognitive differences between different types of procrastinators can provide a scientific basis for effectively addressing and preventing procrastination. Consequently, studying the cognitive differences among various types of procrastinators can offer scientific insights for managing and preventing procrastination.

Numerous studies have shown that procrastinators may have issues with self-regulation, which can lead to excessive mental stress, physical stress responses, fatigue, sleep problems, and other physical and mental health-related issues ([12]; [27]; [42]). This can hinder the development of goal-oriented problem-solving strategies and executive actions, severely affecting work and academic achievements ([43]). Executive function is one of the key cognitive functions of self-regulation ([16]), representing a collection of the various cognitive functions of the brain, including inhibition (inhibitory control), switching (cognitive flexibility), and updating (working memory). These sub-functions work together to help individuals plan, initiate, and complete tasks. For procrastinators, deficits in executive function may lead to difficulties in task initiation, attentional distractions, and an inability to effectively manage time and resources, thereby exacerbating procrastination ([47]). Research has shown that procrastinators exhibit deficiencies in inhibition function ([13]; [30]; [31]), and both switching function and updating function can predict procrastination levels in college students ([30]). Studies by Wang Xuxiang and others have found that both active and passive procrastinators exhibit executive function deficits, with switching function deficits possibly being a key difference between active and passive procrastinators ([43]). Therefore, interventions for procrastination behavior can focus on deficits in executive functions.

Exercise, as a specific form of physical activity, can promote both the physical and mental health of individuals and also change their behavioral habits and patterns. Research has found that individuals who engage in sufficient physical activity (more than 150 min per week) are less likely to demonstrate procrastination behavior ([6]), while less active students exhibit higher levels of procrastination behavior ([46]). In addition to the duration of physical activity, the intensity level of physical activity also affects the degree of procrastination. Shi et al. tested different levels of physical activity intensity (light, moderate, and high) and found that high-intensity physical activity reduces procrastination ([37]). Zhong and Chu also reported that individuals who engage in moderate- or high-intensity physical activity have significantly lower levels of procrastination than those who engage in low-intensity activity ([46]). It is evident that there is a significant negative correlation between physical activity and procrastination behavior. However, further experimental research is needed to investigate the impact of exercise (a specialized form of physical activity) on procrastination behavior among college students, particularly in relation to different types of procrastination.

To date, most studies on the important role of physical activity in reducing procrastination have focused on aspects such as physical activity, self-efficacy, and self-control ([21]; [32]; [48]), and the role of executive function has not yet been fully recognized. Numerous studies have confirmed that exercise is an effective means to enhance and improve executive functions, with consistent findings suggesting that moderate-intensity aerobic exercise is the most beneficial for improving executive functions ([2]; [23]). Moderate-intensity aerobic exercise has been shown to effectively enhance adults’ inhibition ([44]) and switching functions ([15]) and also has a positive impact on updating function ([22]; [29]). Previous research has shown that procrastinators often exhibit one or more deficiencies in their executive sub-functions, and different types of procrastinators may differ in their executive sub-functions. Given the central role of executive function in procrastination behavior and the potential benefits of physical exercise, clarifying how physical exercise improves executive function across different types of procrastinators can provide a scientific basis for developing personalized intervention programs. Furthermore, few studies have explored exercise interventions aimed at improving executive function among procrastinating college students, particularly in the context of school-based physical activity environments. Therefore, to fully understand the impact of physical exercise on executive function in various procrastination types, the following issues need to be addressed: (1) Can enhanced physical exercise reduce procrastination by improving executive functions? (2) What are the differences in the effects of regular exercise on executive functions in college students with different types of procrastination? To explore these questions, this study investigated the effects of moderate-intensity aerobic exercise on the three sub-components of executive function (updating, inhibition, and switching) in college students with different types of procrastination.

## 2. Materials and Methods

### 2.1. Participants’ Characteristics

This study selected students from several classes of public physical education courses in the first and second years of a university in Xi’an as the survey subjects. During theoretical class time, students were asked to complete two questionnaires: the General Procrastination Scale (GPS) and the New Active Procrastination Scale (NAPS). A total of 455 questionnaires were distributed and all were returned and valid. Based on the scoring criteria ([10]), the participants were categorized as passive procrastinators (GPS average score > 3 and NAPS average score > 4), totaling 67 students, active procrastinators (GPS average score > 3 and NAPS average score ≤ 4), totaling 86 students, and non-procrastinators (GPS average score ≤ 3), totaling 302 students.

A further assessment was conducted by phone recruitment, and ultimately, 62 passive procrastinators (12 male, 50 female, 18.65 ± 0.33 years old), 64 active procrastinators (12 male, 52 female, 19.64 ± 1.01 years old), and 64 non-procrastinators (12 male, 52 female, 19.75 ± 1.24 years old) were invited to participate. Based on the principle of having equal numbers of males in each group, the participants were randomly assigned according to different exercise intervention periods as follows: the passive procrastination group (29 students), the passive procrastination exercise group (33 students), the active procrastination group (32 students), the active procrastination exercise group (32 students), the non-procrastination group (32 students), and the non-procrastination exercise group (32 students). The inclusion criteria were as follows: (1) no history of cardiovascular and cerebrovascular disease, neurological disorders, or genetic diseases; (2) normal or corrected vision, with no color blindness or color weakness; (3) normal intelligence and familiarity with basic computer operations; (4) right-handed; (5) no regular exercise habits; and (6) no participation in similar studies.

### 2.2. Experimental Design

A mixed experimental design was used: 3 (procrastination type: active procrastination, passive procrastination, non-procrastination) × 2 (exercise intervention: with exercise, without exercise) × 2 (measurement time: pre-test, post-test). This design allowed us to simultaneously investigate the effects of multiple factors and their interactions on the dependent variable. Procrastination type, measurement time, and exercise intervention were between-subject variables, and the reaction time and accuracy of executive function (inhibition, updating, and switching) were the dependent variables.

### 2.3. Research Methods

#### 2.3.1. Exercise Intervention Program

The exercise intervention program was designed according to the domestic standards for “physical activity levels” ([3]). The specific exercise plan for the exercise group was as follows: The exercise frequency was 3 times per week, with the exercise intensity set at 60–69% of the maximum heart rate (220—age). Each exercise session lasted 30 min, and the activity chosen was running. The exercise intervention lasted for 8 weeks. During the first week, participants were required to engage in low-intensity adaptive training (45–59% of the maximum heart rate (220—age). From the second week onward, the participants performed the exercises at the prescribed intensity. Before and after each exercise session, the participants were required to perform warm-up and cool-down activities. The control group did not receive any exercise intervention and maintained their normal lifestyle and academic routines without engaging in regular physical exercise for the 8 weeks, and the participants’ daily physical activity levels were assessed through weekly verbal inquiries to monitor their exercise habits.

#### 2.3.2. Executive Function Testing

The testing of the three sub-functions of executive function is conducted on a laptop (14-inch screen with a resolution of 1366 × 768). All programs are written using the E-prime 2.0 system. The evaluation metrics are reaction time and accuracy rate. Lower reaction time test values and higher accuracy rates indicate the better performance of the function. The specific tasks for each of the three sub-functions are as follows:

Inhibition Function—Stroop Task ([28]): Studies of the reliability and validity of the instrument show it is a good tool to measure inhibitory cognition with three increasingly complex tasks (congruent, neutral, and incongruent), showing a high intra-class correlation index for different temporal measures (>0.90) ([33]). The experimental materials consisted of four colors (“Red”, “Yellow”, “Blue”, “Green”) and four words (“Red”, “Green”, “Yellow”, “Blue”), which were randomly combined. Participants were required to quickly respond by pressing a key corresponding to the color of the word presented (e.g., red = “D”, yellow = “F”, blue = “J”, green = “K”). The Stroop paradigm had two conditions: congruent (the color and the word meaning are the same) and incongruent (the color and the word meaning are different). The participant’s performance was measured by their inhibition reaction time (incongruent reaction time minus congruent reaction time) and inhibition accuracy (congruent accuracy minus incongruent accuracy).

Switching Function—More-odd Shifting Task ([35]): The task-switching paradigm offers enormous possibilities to study cognitive control with good reliability and validity ([19]). The experimental materials consisted of the numbers 1–9 (excluding 5). There were three types of judgments required: a. When black numbers were presented, participants were required to judge whether the number was smaller or larger than 5. If smaller, they pressed “F”; if larger, they pressed “J”. b. When green numbers were presented, participants were required to judge whether the number was odd or even. If odd, they pressed “F”; if even, they pressed “J”. c. When black or green numbers were presented, participants were required to switch between size and parity judgments and respond accordingly. The More-odd shifting paradigm was assessed by switching reaction time (switching reaction time − [(size reaction time + odd/even reaction time)/2]) and switching accuracy ([(size accuracy + odd/even accuracy)/2] − switching accuracy).

Updating Function—N-back Task ([39]): The N-back task is a widely used standard measure of executive working memory in cognitive neuroscience with good reliability and validity ([17]). The experimental materials consisted of 26 English letters. There were two types of judgments: a. 1-back task: Starting from the second letter, participants were required to judge whether the current letter was the same as the previous letter. If it was the same, they pressed the “spacebar”; if different, they did nothing. b. 2-back task: Starting from the third letter, participants were required to judge whether the current letter was the same as the second-to-last letter. If it was the same, they pressed the “spacebar”; if different, they did nothing. The N-back paradigm was assessed by updating reaction time (2-back reaction time minus 1-back reaction time) and updating accuracy (1-back accuracy minus 2-back accuracy).

Prior to the testing of each task, the test subject was instructed by the main examiner to practice the task until the test subject was no less than 85% correct, and then the formal testing began.

### 2.4. Data Analysis

Statistical analysis was performed using SPSS 22.0 software. The data were presented as mean ± standard deviation (M ± SD). A one-way ANOVA was used to analyze the developmental characteristics of executive functions among college students with different types of procrastination. Additionally, an independent sample t-test was used to test the homogeneity of executive function levels across the different groups before exercise. Three-way repeated-measures ANOVA was applied to analyze the effects of procrastination type, exercise intervention, and their interaction on executive functions (inhibition, switching, and updating). The calculations of the Sum of Squares and the degrees of freedom were used to evaluate the goodness of model fit and to conduct hypothesis testing. If the effects were significant, post-hoc comparisons were conducted using the Bonferroni method. The significance level for the statistical analysis was set at *p* < 0.05. For the reaction time analysis, data outside of M ± 3 SD were excluded.

## 3. Results

### 3.1. Analysis of Executive Function Sub-Skills in College Students with Different Types of Procrastination

A one-way ANOVA analysis was conducted on the executive function of college students with active procrastination, passive procrastination, and non-procrastination to examine the developmental characteristics of executive functions in college students with different procrastination types. The results in Table 1 show that college students with active procrastination had higher inhibition, switching, and updating scores compared to college students with non-procrastination, while college students with passive procrastination had lower inhibition, switching, and updating scores than those with non-procrastination. Significant differences were found between active procrastination and passive procrastination college students in their inhibition scores and accuracy, as well as their updating scores and accuracy *(p* < 0.05). In addition, the inhibition accuracy of college students with non-procrastination was significantly higher than that of college students with passive procrastination (*p* < 0.05), and the updating accuracy of the active procrastination students was significantly higher than that of the college students with non-procrastination (*p* < 0.05).

### 3.2. Homogeneity Test for Pre-Test Executive Function

Descriptive statistics of executive function before and after the exercise intervention in students with different types of procrastination are presented in Table 2.

Independent sample *t*-tests were performed on the pre-test scores of the inhibition, switching, and updating functions between the exercise group and the control group for college students with active procrastination, passive procrastination, and non-procrastination.

For the active procrastination college students, there were no significant differences in inhibition function (inhibition score: *t* = −0.345, *p* = 0.733; inhibition accuracy: *t* = −0.934, *p* = 0.361), switching function (switching score: *t* = 0.833, *p* = 0.414; switching accuracy: *t* = −1.675, *p* = 0.110), and updating function (updating score: *t* = 1.141, *p* = 0.267; updating accuracy: *t* = 0.098, *p* = 0.923), indicating that the levels of executive function were homogeneous before the intervention.

For the passive procrastination college students, there were no significant differences in inhibition function (inhibition score: *t* = 0.031, *p* = 0.975; inhibition accuracy: *t* = 1.300, *p* = 0.205), switching function (switching score: *t* = −1.001, *p* = 0.325; switching accuracy: *t* = 0.260, *p* = 0.797), and updating function (updating score: *t* = 0.032, *p* = 0.975; updating accuracy: *t* = −0.225, *p* = 0.824), indicating that the levels of executive function were homogeneous before the intervention.

Similarly, the for non-procrastination students, there were no significant differences in the pre-test scores in inhibition function (inhibition score: *t* = 0.093, *p* = 0.926; inhibition accuracy: *t* = 0.519, *p* = 0.609), switching function (switching score: *t* = −1.628, *p* = 0.117; switching accuracy: *t* = −0.150, *p* = 0.882), and updating function (updating score: *t* = −1.092, *p* = 0.286; updating accuracy: *t* = −0.672, *p* = 0.508), indicating homogeneity between the exercise and control groups before the intervention.

### 3.3. Impact of Aerobic Exercise Intervention on Executive Function in Students with Different Types of Procrastination

Three-way repeated-measures ANOVA was used to analyze the effects of the aerobic exercise intervention on executive functions in college students with active procrastination, passive procrastination, and non-procrastination (Table 3).

#### 3.3.1. Interference Inhibition Task—Stroop Task

Repeated-measures ANOVA was performed on the reaction time and accuracy of the Stroop task to examine the intervention effects. The results of the intervention effects on the Stroop task and the other tasks are shown in Figure 1.

For reaction time, the main effect of procrastination type was significant, F(2, 60) = 3.953, *p* = 0.034, η_p_^2^ = 0.264. Post-hoc comparisons revealed that college students with passive procrastination had shorter reaction times than those with active procrastination. The interaction between time and group was significant, F(1, 28) = 6.667, *p* = 0.025, η_p_^2^ = 0.377, and simple effects analysis showed that, compared to the non-procrastination control group, the reaction speed of the non-procrastination exercise group improved after the intervention, F(1, 28) = 6.128, *p* = 0.031, η_p_^2^ = 0.358, while the other five groups did not show an improvement.

For accuracy, the interaction between group and type was significant: F(2, 28) = 4.895, *p* = 0.033, η_p_^2^ = 0.495. Simple effects analysis revealed that, after the exercise intervention, the inhibition accuracy of the non-procrastination control group was significantly different from that of the non-procrastination exercise group: F(1, 28) = 5.348, *p* = 0.041, η_p_^2^ = 0.327. The passive procrastination college students in the exercise group also showed a significant increase in inhibition accuracy compared to the control group: F(1, 28) = 5.101, *p* = 0.043, η_p_^2^ = 0.298.

#### 3.3.2. Cognitive Flexibility Task—More-Odd Shifting Task

A three-way repeated-measures ANOVA was conducted on the reaction times from the More-odd shifting task and found that there were no significant interactions between measurement time, type, and group. However, the main effect of measurement time was significant: F(1, 28) = 75.149, *p* < 0.001, η_p_^2^ = 0.883. Further post-hoc comparisons revealed that the non-procrastination type college students in the exercise group showed no difference in switching reaction times before and after the intervention, meaning that these students did not exhibit any effects on switching function due to the training. In contrast, for the other five groups of college students, their switching reaction times at the end of the experiment were significantly improved compared to before the experiment (*p*-values were 0.000, 0.007, 0.048, 0.004, 0.002; η_p_^2^ values were 0.730, 0.504, 0.200, 0.521, 0.546, respectively).

For accuracy, the interaction between type and time was significant: F(2, 28) = 7.332, *p* = 0.024, η_p_^2^ = 0.710. Simple effects analysis revealed that, by fixing the subject types to observe differences between the testing time, the control group of non-procrastinating college students showed a significant difference in switching accuracy before and after the intervention: F(1, 28) = 6.162, *p* = 0.027, η_p_^2^ = 0.322; and the exercise group of passive procrastinating college students significantly improved in switching accuracy before and after the intervention: F(1, 28) = 7.917, *p* = 0.017, η_p_^2^ = 0.419. By fixing the testing time to observe differences between the subject types, the results showed that at the end of the intervention, there was a significant difference in switching accuracy among college students of different procrastination types in the control group: F(2, 60) = 4.896, *p* = 0.019, η_p_^2^ = 0.329. Pairwise comparisons found that the switching accuracy of the non-procrastinating control group college students was significantly higher than that of the passive procrastinating control group college students (*p* = 0.022, 95% CI: [−0.160, −0.01374]).

#### 3.3.3. Working Memory Update Task—N-Back Task

The results of the updating function test showed that in terms of reaction time, the main effect of group was highly significant, F(1, 28) = 8.131, *p* = 0.019, η_p_^2^ = 0.475, indicating that there was a significant difference in the updating function between the control and the exercise group of college students. The interaction between group and time was also significant, F(1, 28) = 7.544, *p* = 0.023, η_p_^2^ = 0.456, indicating that there were significant differences in the changes in updating function before and after the test among different groups of college students. Further simple effects tests revealed that by fixing the testing time to observe differences between the groups, the post-intervention tests revealed that the exercise group of active procrastination college students exhibited significantly shorter updating reaction times compared to the control group of active procrastination college students: F(1, 28) = 43.250, *p* < 0.001, η_p_^2^ = 0.797. By fixing the group constant to observe differences between the testing times, the updating reaction time of the active procrastinators in the exercise group was significantly shorter in the post-test than in the pre-test: F(1, 28) = 5.309, *p* = 0.042, η_p_^2^ = 0.326; and the updating reaction time of non-procrastinators in the exercise group was also significantly shorter in the post-test than in the pre-test: F(1, 28) = 4.958, *p* = 0.048, η_p_^2^ = 0.311.

For accuracy, the interaction between type, time, and group was significant: F(2, 28) = 7.986, *p* = 0.012, η_p_^2^ = 0.666. Further simple effects analysis revealed that there were no significant simple two-way interactions.

## 4. Discussion and Analysis

This study examined the differences in executive function among college students with different types of procrastination, as well as the effects of exercise on improving executive function in these students. The results indicate that the key difference between active procrastination and passive procrastination is in the inhibition and updating functions, with passive procrastinators showing significant deficiencies in inhibition accuracy. After eight weeks of exercise training, aerobic exercise was found to significantly improve inhibition reaction times and updating reaction times in non-procrastinators, significantly improve switching accuracy and inhibition accuracy in passive procrastinators, and significantly improve updating reaction times in active procrastinators.

### 4.1. Differences in Executive Function Among College Students with Different Types of Procrastination

The most fundamental difference between active procrastination and passive procrastination lies in whether the behavior is a rational tendency. Active procrastination has a minimal negative impact on individuals. In this study, college students with active procrastination demonstrated superior executive function, with significantly better inhibition and updating abilities than passive procrastinators, and a higher updating accuracy than non-procrastinators. Chu and Choi emphasize that active procrastination is a voluntary choice made after careful consideration, where procrastinators have good self-control, can effectively manage their time, and ensure tasks are completed accurately and efficiently ([5]). Procrastination is seen as an effective strategy to avoid impulsiveness. Similarly, Lu Cuiyan et al. also consider active procrastination as an active behavior, where individuals can motivate themselves to reach their full potential under time pressures ([25]). Passive procrastination, on the other hand, is a negative form of procrastination. This study confirmed some of Wang Xuxiang et al.’s findings, indicating that passive procrastinators exhibit deficiencies in their inhibition and updating functions ([43]). Furthermore, we also found that active procrastinator college students showed a significantly lower inhibition accuracy than non-procrastinator students, which corroborates the conclusions of Rabin et al. ([30]), Gustavson et al. ([31]), and Rebetez et al. ([13]). These studies suggest that, compared to active procrastinators and non-procrastinators, passive procrastinators lack sufficient cognitive control, leading to poorer executive function. Moreover, according to Gustavson et al.’s analysis, the switching and working memory functions are unrelated to procrastination ([13]). Our results support this by showing no statistical difference in the switching sub-function between procrastinator and non-procrastinator college students.

### 4.2. Effects of Exercise on Different Sub-Functions of Executive Function in College Students with Different Types of Procrastination

The inhibition function is a core component of executive function. Previous behavioral and neuroimaging studies have suggested that procrastination behavior is generally caused by a failure of self-regulation or by avoidance coping strategies ([1]; [21]). Compared to non-procrastinators, procrastinators tend to have weaker cognitive control leading to more impulsive behaviors. Numerous studies have found that exercise can have a positive effect on inhibition control. For example, Kamijo et al. found that moderate-intensity acute aerobic exercise (cycling) significantly improved inhibition control abilities and individual attention ([18]). Yang Yongtao et al.’s study also found that moderate-intensity sustained aerobic exercise (stair climbing) significantly enhanced inhibition control abilities in college students ([45]). In this study, after eight weeks of aerobic exercise intervention (running), we found that exercise significantly shortened inhibition reaction times in non-procrastinator college students, and significantly improved inhibition accuracy in active procrastinator college students.

In terms of behavior, both updating and inhibition require the suppression of irrelevant information ([26]). Some studies have found that working memory updating can significantly predict procrastination behavior in college students ([30]). Gai Xiaosong et al. found that after one-time training, the intensity of exercise in body-sensory games improved children’s executive function, especially their working memory ([11]). In our study, after the exercise intervention, both non-procrastinators and active procrastinator college students showed significant improvements in updating reaction times. Active procrastination can be seen as a form of proactive procrastination, allowing individuals to develop more effective strategies or plans, and our results support the notion that exercise enhances updating function. However, we did not find improvements in the updating function of passive procrastinator college students, which may be related to the high correlation between working memory and self-control. In addition, individual baseline cognitive abilities (such as working memory and executive function) and motivation levels may be important confounding variables that affect their response to the exercise interventions. For example, individuals with higher cognitive abilities and highly motivated students may benefit more from exercise because they have richer cognitive resources, participate more actively in exercise interventions, and can better integrate the positive effects of exercise ([40]). Conversely, students with lower baseline cognitive abilities and low motivation may show less improvement in exercise interventions because they have limited cognitive resources, lower interest in exercise interventions, and insufficient participation, leading to less effective intervention outcomes ([14]; [34]). Colcombe et al. noted in their meta-analysis that aerobic exercise has selective effects on executive function, and different types of exercise have varying impacts on executive functions ([7]).

Switching ability involves using competing cognitive resources between two tasks and is the individual’s ability to shift between different perspectives to adapt to changing environmental demands ([9]). Compared to updating function, switching function requires more cognitive resources, and therapy reflects cognitive processing advantages. Studies have found that 24 weeks of soccer training made the experimental group’s switching reaction time faster ([8]), and 8 weeks of swimming training improved children’s cognitive flexibility ([38]). Moreover, exercise can promote the release of dopamine neurotransmitters, which regulate cognitive flexibility and influence switching ability ([20]), and moderate-intensity acute aerobic exercise can significantly enhance individual focus ([24]). In our study, after the exercise intervention, passive procrastinator college students showed significant improvement in switching ability, likely due to both the exercise itself and the task demands of the exercise.

## 5. Conclusions

In summary, this study employed a standard experimental paradigm to systematically and comprehensively examine the effects of aerobic exercise on the executive function of college students with different types of procrastination. It was found that significant differences exist in the inhibition and updating functions between active procrastinators and passive procrastinators, with passive procrastinators exhibiting notable deficiencies in inhibition accuracy. Furthermore, aerobic exercise significantly enhances the inhibition and updating sub-functions in non-procrastinating college students. Additionally, the aerobic exercise intervention positively improves updating reaction times in active procrastinators and enhances the inhibition accuracy and switching ability in passive procrastinators. These results demonstrate that aerobic exercise can effectively promote the development of executive function in college students, with varying effects depending on the type of procrastination. This finding provides a new approach for targeted exercise interventions tailored to different types of procrastinators and carries significant theoretical and practical implications.

Meanwhile, it is necessary to recognize that this study has some limitations that should be addressed in future research. Firstly, the intervention duration (8 weeks) for different types of procrastinators may not capture the long-term effects, and the cumulative effects of the intervention remain unclear. In follow-up studies, we will further observe the sustainability of the intervention effects. Secondly, self-reported measures (e.g., the General Procrastination Scale) may introduce potential biases. Future research could incorporate more objective measurement methods. Thirdly, future research needs to further explore other factors (such as gender, cultural differences, and coping strategies) that influence the effects of exercise on executive function in different types of procrastinators. This will help to strictly control the relevant variables and ensure the rigor and scientific validity of experiments. Finally, it is necessary to consider incorporating other forms of exercise (such as resistance training or mindfulness-based physical activities) to identify effective exercise interventions for improving procrastination behaviors.

## Figures and Tables

**Figure 1 behavsci-15-00225-f001:**
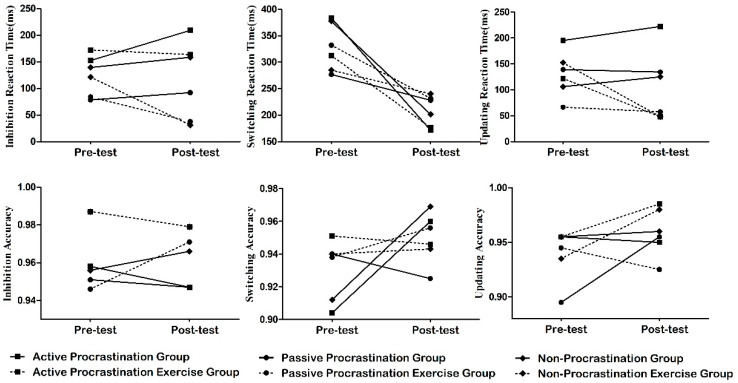
Changes in the executive function trends of college students with different types of procrastination before and after the eight week exercise intervention. Inhibition sub-function: assessed through the Stroop task by calculating the inhibitory reaction time and inhibitory accuracy. Switching sub-function: assessed through the More-odd shifting task by calculating the switching reaction time and switching accuracy. Updating sub-function: assessed through the N-back task by calculating the updating reaction time and updating accuracy.

**Table 1 behavsci-15-00225-t001:** Comparison of executive function scores among college students with different types of procrastination (M ± SD).

Function		Active Procrastination Group (*n* = 64)	Passive Procrastination Group (*n* = 62)	Non-Procrastination Group (*n* = 64)	F-Value	*p*-Value
Inhibition	Inhibition Reaction Time (ms)	184.941 ± 120.023	85.462 ± 186.645	113.23 ± 90.56	3.167	0.048
	Inhibition Accuracy	0.966 ± 0.041	0.944 ± 0.031	0.972 ± 0.032	4.854	0.010
Switching	Switching Reaction Time (ms)	344.125 ± 135.564	296.766 ± 148.587	326.308 ± 171.582	0.692	0.504
	Switching Accuracy	0.930 ± 0.028	0.915 ± 0.048	0.937 ± 0.044	1.884	0.159
Updating	Updating Reaction Time (ms)	159.391 ± 122.593	90.871 ± 118.419	119.846 ± 117.128	2.279	0.120
	Updating Accuracy	0.963 ± 0.028	0.934 ± 0.044	0.929 ± 0.046	3.823	0.026

**Table 2 behavsci-15-00225-t002:** Descriptive statistics of executive function at two measurement time points (M ± SD).

Executive Function		Active Procrastination Group (*n* = 32)	Active Procrastination Exercise Group (*n* = 32)	Passive Procrastination Group (*n* = 29)	Passive Procrastination Exercise Group (*n* = 33)	Non-Procrastination Group (*n* = 32)	Non-Procrastination Exercise Group (*n* = 32)
Inhibition Reaction Time (ms)	Pre-test	152.645 ± 135.804	172.405 ± 144.608	78.411 ± 186.671	84.114 ± 189.360	139.524 ± 94.195	121.425 ± 117.032
	Post-test	209.327 ± 159.818	163.681 ± 128.356	92.247 ± 181.933	37.653 ± 133.374	158.670 ± 104.918	31.159 ± 138.225
Inhibition Accuracy	Pre-test	0.958 ± 0.047	0.987 ± 0.018	0.951 ± 0.018	0.946 ± 0.016	0.987 ± 0.015	0.956 ± 0.053
	Post-test	0.947 ± 0.050	0.979 ± 0.016	0.947 ± 0.051	0.981 ± 0.012	0.979 ± 0.013	0.966 ± 0.018
Switching Reaction Time (ms)	Pre-test	383.591 ± 142.446	312.318 ± 127.358	276.909 ± 182.227	332.227 ± 127.537	377.636 ± 184.667	284.818 ± 158.102
	Post-test	171.590 ± 121.433	176.409 ± 63.889	227.954 ± 167.743	231.727 ± 90.838	201.636 ± 150.180	240.227 ± 119.209
Switching Accuracy	Pre-test	0.904 ± 0.016	0.951 ± 0.030	0.940 ± 0.034	0.938 ± 0.031	0.912 ± 0.087	0.940 ± 0.013
	Post-test	0.960 ± 0.013	0.946 ± 0.054	0.925 ± 0.034	0.956 ± 0.023	0.969 ± 0.022	0.943 ± 0.027
Updating Reaction Time (ms)	Pre-test	195.000 ± 144.230	121.700 ± 94.218	139.100 ± 102.952	66.300 ± 118.373	106.100 ± 145.914	152.100 ± 110.321
	Post-test	222.200 ± 101.808	47.900 ± 112.838	134.300 ± 70.419	57.400 ± 111.904	124.800 ± 124.144	48.100 ± 90.478
Updating Accuracy	Pre-test	0.955 ± 0.021	0.955 ± 0.033	0.895 ± 0.041	0.945 ± 0.045	0.955 ± 0.033	0.935 ± 0.055
	Post-test	0.950 ± 0.040	0.985 ± 0.014	0.955 ± 0.027	0.925 ± 0.061	0.960 ± 0.038	0.980 ± 0.021

**Table 3 behavsci-15-00225-t003:** Overview of variance analysis on the impact of the exercise intervention on executive function in students with different types of procrastination.

		Type III Sum of Squares	*df*	Mean Square	*F*-Value	*p*-Value
Inhibition Reaction Time	Group	48,570.467	1	48,570.467	1.435	0.256
	Type	250,761.949	2	125,380.975	3.953	0.034 *
	Time	3112.273	1	3112.273	0.153	0.703
	Group × Type	24,217.405	2	12,108.702	0.590	0.563
	Group × Time	55,278.711	1	55,278.711	6.667	0.025 *
	Type × Time	22,154.692	2	11,077.346	0.527	0.597
	Group × Type × Time	4374.861	2	2187.431	0.125	0.883
Inhibition Accuracy	Group	0.001	1	0.001	1.985	0.218
	Type	0.003	2	0.002	1.089	0.373
	Time	0.000	1	0.000	0.129	0.734
	Group × Type	0.009	2	0.004	4.895	0.033 *
	Group × Time	0.002	1	0.002	1.194	0.324
	Type × Time	0.002	2	0.001	0.976	0.410
	Group × Type × Time	0.001	2	0.000	0.689	0.524
Switching Reaction Time	Group	3477.320	1	3477.320	0.128	0.728
	Type	5069.140	2	2534.570	0.069	0.934
	Time	472,503.835	1	472,503.835	75.149	0.000 **
	Group × Type	26,356.163	2	13,178.081	0.519	0.603
	Group × Time	22,295.002	1	22,295.002	1.925	0.195
	Type × Time	55,599.966	2	27,799.983	2.402	0.116
	Group × Type × Time	48,421	2	24,210.775	2.340	0.122
Switching Accuracy	Group	0.001	1	0.001	0.526	0.521
	Type	1.517 × 10^−5^	2	7.583 × 10^−6^	0.004	0.996
	Time	0.004	1	0.004	2.341	0.223
	Group × Type	0.001	2	0.000	0.297	0.753
	Group × Time	0.002	1	0.002	1.289	0.339
	Type × Time	0.002	2	0.001	7.332	0.024 *
	Group × Type × Time	0.006	2	0.003	2.329	0.178
Updating Reaction Time	Group	152,653.333	1	152,653.333	80,131	0.019 *
	Type	51,153.817	2	25,576.908	1.782	0.197
	Time	17,666.133	1	17,666.133	1.284	0.286
	Group × Type	58,992.517	2	29,496.258	2.197	0.140
	Group × Time	43,244.033	1	43,244.033	7.544	0.023 *
	Type × Time	6422.217	2	3211.108	0.374	0.693
	Group × Type × Time	19,938.717	2	9969.358	0.782	0.472
Updating Accuracy	Group	0.001	1	0.001	1.228	0.330
	Type	0.012	2	0.006	1.862	0.217
	Time	0.006	1	0.006	5.781	0.074
	Group × Type	0.001	2	0.000	0.168	0.849
	Group × Time	1.042 × 10^−5^	1	1.042 × 10^−5^	0.026	0.880
	Type × Time	0.000	2	0.000	0.161	0.854
	Group × Type × Time	0.012	2	0.006	7.986	0.012 *

Note: * *p* < 0.05, ** *p* < 0.01.

## Data Availability

The datasets used and analyzed during the current study are available from the corresponding author on reasonable request.

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
