# Peer review of "The Effects of Aerobic Exercise on Executive Function: A Comparative Study Among Active, Passive, and Non-Procrastinating College Students"

_behavsci, 2025, doi:10.3390/bs15020225_

Round 1

Reviewer 1 Report

Comments and Suggestions for Authors

Review of the Manuscript: “An Experimental Study on the Effects of Aerobic Exercise on Executive Function in College Students with Different Procrastination Types”

General Comments:

The manuscript addresses a timely and relevant topic concerning the impact of aerobic exercise on executive function among college students with varying procrastination types. This study offers significant contributions to understanding the differential effects of exercise interventions on cognitive and behavioral outcomes. While the manuscript is well-structured and supported by empirical evidence, certain areas require further clarity, refinement, and elaboration to enhance its overall quality and readability.

Specific Comments:

Title:

·       The title accurately reflects the content of the study but could benefit from more specificity. For instance:

*Suggested revision:* “Effects of Aerobic Exercise on Executive Function: A Comparative Study Among Active, Passive, and Non-Procrastinating College Students.”

Abstract:

·       The abstract provides an adequate overview but contains overly technical terms that may not be accessible to a broader audience. Simplify the language to enhance readability.

·       Key numerical results (e.g., reaction times, significance levels) are presented but lack context. Explain what these results mean in practical terms.

·       Consider rephrasing the conclusion to emphasize broader implications for educational or health interventions.

Introduction:

·       The introduction provides a robust theoretical background but has areas of redundancy. For example, the definitions of procrastination and its types are repeated.

·       Clearly articulate the research gap:

- While the study addresses the role of executive functions in procrastination, it could better highlight why these functions are critical for intervention strategies.

·       Provide more recent references where applicable (e.g., studies published post-2020).

Materials and Methods:

·       **Participants:**

- Inclusion criteria are detailed; however, the rationale for selecting 'right-handed' participants could be clarified.

- Specify whether gender differences were considered, as they may influence executive function and procrastination behaviors.

·       **Experimental Design:**

- The 3×2×2 design is appropriate but would benefit from a brief explanation for readers unfamiliar with factorial designs.

- Clearly describe the procedures for randomization and group allocation.

·       **Exercise Protocol:**

- While the protocol is well-described, include details about participants’ compliance and how it was monitored.

- Discuss whether the control group’s activities were supervised to ensure validity.

·       **Executive Function Testing:**

- The Stroop, More-Odd Shifting, and N-back tasks are appropriate measures. However, explain why these tasks were chosen over other executive function tests.

- Provide references for the psychometric properties of these tests.

Results:

·       **Descriptive Statistics:**

- Include more detailed explanations of the data trends observed in Table 2 (e.g., why inhibition accuracy improved in specific groups).

·       **Inferential Statistics:**

- The repeated-measures ANOVA results are comprehensive. However, ensure all significant interactions are accompanied by effect sizes (e.g., Cohen’s d or partial eta squared).

- Clarify the practical implications of significant results. For instance, how do changes in reaction times translate to real-world benefits for students?

·       **Figures and Tables:**

- Enhance the readability of tables by standardizing formatting (e.g., consistent decimal places).

- Ensure figure captions provide sufficient detail to be understood independently of the text.

Discussion:

·       The discussion effectively interprets the results but requires deeper engagement with the literature.

- Compare findings with previous studies on procrastination and executive function. Are the improvements observed consistent with earlier interventions?

- Address potential confounding variables (e.g., baseline cognitive abilities, motivational differences).

·       Expand on practical applications:

- How can universities implement aerobic exercise programs to reduce procrastination and improve academic outcomes?

·       Acknowledge limitations:

- The study duration (8 weeks) may not capture long-term effects. Mention the need for follow-up studies.

- Discuss potential biases due to self-reported measures (e.g., the General Procrastination Scale).

Conclusion:

·       The conclusion summarizes the findings well but should include specific recommendations for future research, such as:

- Exploring gender or cultural differences.

- Testing other forms of exercise (e.g., resistance training or mindfulness-based physical activity).

References:

·        Replace older references where newer studies are available to strengthen the manuscript’s relevance.

Minor Comments:

·       1. Standardize terminology:

·       - Use either 'active procrastination' or 'proactive procrastination' consistently.

·       - Avoid switching between 'exercise group' and 'intervention group.'

·       2. Grammar and style:

·       - Correct typographical errors (e.g., '8-weeks' should be '8 weeks').

·       - Use active voice where possible to improve readability.

·       3. Figures:

·       - Improve figure resolution for better visual clarity.

Author Response

Reviewer 1

(1) Title:The title accurately reflects the content of the study but could benefit from more specificity. For instance:

*Suggested revision:* “Effects of Aerobic Exercise on Executive Function: A Comparative Study Among Active, Passive, and Non-Procrastinating College Students.”

Response 1- (1) 

  According to the advice of reviewer, we have revised the title in the manuscript (Lines 1-3).

(2) Abstract:

  • The abstract provides an adequate overview but contains overly technical terms that may not be accessible to a broader audience. Simplify the language to enhance readability.

Response 1- (2) 

We thank the reviewer for their valuable suggestions. However, simplifying the language to enhance readability in this section might alter the meaning of the technical terms, so we have chosen not to make changes here.

(3) Abstract:

  • Key numerical results (e.g., reaction times, significance levels) are presented but lack context. Explain what these results mean in practical terms.

Response 1- (3) 

According to the advice of reviewer, we have provided a detailed explanation of the practical implications of the results (Lines 38-41, Lines 46-50).

(4) Abstract:

  • Consider rephrasing the conclusion to emphasize broader implications for educational or health interventions.

Response 1- (4) 

According to the advice of reviewer, we have revised the conclusion (Lines 50-53).

(5) Introduction:

  • The introduction provides a robust theoretical background but has areas of redundancy. For example, the definitions of procrastination and its types are repeated.

Response 1- (5) 

According to the advice of reviewer, we have revised the content of this paragraph. (Lines 57-59, Lines 62-72).

(6) Introduction:

  • Clearly articulate the research gap: While the study addresses the role of executive functions in procrastination, it could better highlight why these functions are critical for intervention strategies.

Response 1- (6) 

According to the advice of reviewer, we have revised this paragraph. (Lines 78-85).

(7) Introduction:

  • Provide more recent references where applicable (e.g., studies published post-2020).

Response 1- (7) 

According to the advice of reviewer, we have updated the references.

(8) Materials and Methods:

  • **Participants:** Inclusion criteria are detailed; however, the rationale for selecting 'right-handed' participants could be clarified.

Response 1- (8) 

In this study, to minimize the impact of variability in cerebral lateralization on the research outcomes, we selected right-handed participants to ensure greater consistency among the subjects.

(9) Materials and Methods:

  • **Participants:** Specify whether gender differences were considered, as they may influence executive function and procrastination behaviors.

Response 1- (9) 

According to the advice of reviewer, we have conducted statistical analyses on the age and gender of the participants. (Lines 149-151).

(10) Materials and Methods:

  • **Experimental Design:** The 3×2×2 design is appropriate but would benefit from a brief explanation for readers unfamiliar with factorial designs.

Response 1- (10) 

According to the advice of reviewer, we have provided an explanation of the significance of this design. (Lines 166-167).

(11) Materials and Methods:

  • **Experimental Design:** Clearly describe the procedures for randomization and group allocation.

Response 1- (11) 

According to the advice of reviewer, we have revised this sentence. (Lines 151-153).

(12) Materials and Methods:

  • **Exercise Protocol:** While the protocol is well-described, include details about participants’ compliance and how it was monitored.

Response 1- (12) 

In this study, we required all participants in the intervention groups to use a fitness application (Keep App) to record each exercise session and upload it to a WeChat group. The researchers of this experiment supervised and guided the exercise process.

(13) Materials and Methods:

  • **Exercise Protocol:**Discuss whether the control group’s activities were supervised to ensure validity.

Response 1- (13) 

According to the advice of reviewer, we have revised this sentence. (Lines 185-186).

(14) Materials and Methods:

  • **Executive Function Testing:** The Stroop, More-Odd Shifting, and N-back tasks are appropriate measures. However, explain why these tasks were chosen over other executive function tests.

Response 1- (14) 

The selection of the Stroop, More-Odd Shifting, and N-back tasks as measures of executive function is based on: (1) These tasks have been widely used in previous research and have demonstrated sensitivity to individual differences in executive function. (2) The cognitive processes measured by these tasks are directly related to the self-regulation challenges commonly faced by individuals with procrastination tendencies. (3) These tasks are well-validated with established psychometric properties, ensuring the reliability and validity of measuring executive function.

(15) Materials and Methods:

  • **Executive Function Testing:** Provide references for the psychometric properties of these tests.

Response 1- (15) 

According to the advice of reviewer, we have added relevant references. (Lines 193-197, 206-208, 219-221).

(16) Results:

  • **Descriptive Statistics:** Include more detailed explanations of the data trends observed in Table 2 (e.g., why inhibition accuracy improved in specific groups).

Response 1- (16) 

In Table 2, we have statistically analyzed the results from the pre-test and post-test, with a detailed analysis of variance provided in section 3.3.1.

(17) Results:

  • **Inferential Statistics:** The repeated-measures ANOVA results are comprehensive. However, ensure all significant interactions are accompanied by effect sizes (e.g., Cohen’s d or partial eta squared).

Response 1- (17) 

According to the advice of reviewer, we have added significant interactions with effect sizes (partial eta squared).

(18) Results:

  • **Inferential Statistics:** Clarify the practical implications of significant results. For instance, how do changes in reaction times translate to real-world benefits for students?

Response 1- (18) 

According to the advice of reviewer, we have added the relevant explanations (Lines 190-192).

(19) Results:

  • **Figures and Tables:** Enhance the readability of tables by standardizing formatting (e.g., consistent decimal places).

Response 1- (19) 

According to the advice of reviewer, we have standardized the data to three decimal places. (Tables 1-3).

(20) Results:

  • **Figures and Tables:** Ensure figure captions provide sufficient detail to be understood independently of the text.

Response 1- (20) 

According to the advice of reviewer, we have revised the figure captions. (Lines306-310).

(21) Discussion:

  • Compare findings with previous studies on procrastination and executive function. Are the improvements observed consistent with earlier interventions?

Response 1- (21) 

In our study, we did not find improvements in the updating function of passive procrastinator college students, which is inconsistent with earlier interventions. (Lines 437-439).

(22) Discussion:

  • Address potential confounding variables (e.g., baseline cognitive abilities, motivational differences).

Response 1- (22) 

According to the advice of reviewer, we have added this section. (Lines 439-449).

(23) Discussion:

  • Acknowledge limitations:

- The study duration (8 weeks) may not capture long-term effects. Mention the need for follow-up studies.

- Discuss potential biases due to self-reported measures (e.g., the General Procrastination Scale).

Response 1- (23) 

According to the advice of reviewer, we have revised this section. (Lines 480-493).

(24) Conclusion:

  • The conclusion summarizes the findings well but should include specific recommendations for future research, such as: Exploring gender or cultural differences.Testing other forms of exercise (e.g., resistance training or mindfulness-based physical activity).

Response 1- (24) 

According to the advice of reviewer, we have revised this section. (Lines 486-493).

(25) References:

  • Replace older references where newer studies are available to strengthen the manuscript’s relevance.

Response 1- (25) 

According to the advice of reviewer, we have replaced some of the references.

Minor Comments:

(1) 1. Standardize terminology:

  • Use either 'active procrastination' or 'proactive procrastination' consistently.
  • Avoid switching between 'exercise group' and 'intervention group.'

Response 1- (1) 

According to the advice of reviewer, we have revised the relevant expressions mentioned above.

(2) 2. Grammar and style:

  • Correct typographical errors (e.g., '8-weeks' should be '8 weeks').
  • Use active voice where possible to improve readability.

Response 1- (2) 

According to the advice of reviewer, we have revised the relevant expressions mentioned above.

(3) 3. Figures:

  • Improve figure resolution for better visual clarity.

Response 1- (3) 

According to the advice of reviewer, we have improved the resolution of the figures.

Reviewer 2 Report

Comments and Suggestions for Authors

1. The Introductory part does not justify the objectives clearly. You say only: "To fully understand the effect of physical exercise on improving the..." Please justify the necessity to have such objectives. 

2. PLease specify how did you control if participants did all the exercises. How many skipped how many meetings?

3. Table 1 has a different reporting than the usual one. Please see APA or other style reporting for tables.

4. p values for Anova and other values could be inserted in the tables

5. Please explain in the title or in a note what the data represent in the tables.

6. The Conclusion section is atipical. It should contain a short summary, limits, further research... I see you put some of these above in Discussion section. 

7. proofread for spaces in the article: F (1, 28)=75.149

8. CI are reported somehow differently: CI: -0.160 — -0.0137

9. How do you explain that in most significance tests with ANOVA you did not get significant differences?

10. In Bibliography, years are written with bold.

Author Response

Reviewer 2

(1) The Introductory part does not justify the objectives clearly. You say only: "To fully understand the effect of physical exercise on improving the..." Please justify the necessity to have such objectives.

Response 2- (1)

According to the reviewer’s suggestion, we have added relevant information in the introduction sections. (Lines 122-130).

(2) Please specify how did you control if participants did all the exercises. How many skipped how many meetings?

Response 2- (2)

In this study, we required all participants in the intervention groups to use a fitness application (Keep App) to record each exercise session and upload it to a WeChat group. The researchers of this experiment supervised and guided the exercise process. The statistical results indicate that two female participants missed 2 and 3 exercise sessions respectively due to physiological reasons.

(3) Table 1 has a different reporting than the usual one. Please see APA or other style reporting for tables.

  Response 2- (3)

According to the advice of reviewer, we have revised the Table 1.

(4) p values for Anova and other values could be inserted in the tables

Response 2- (4)

According to the advice of reviewer, we have added the p value to Table 1.

(5) Please explain in the title or in a note what the data represent in the tables.

Response 2- (5)

According to the advice of reviewer, we have added relevant information to the table 1 and 2 headers.

(6) The Conclusion section is atipical. It should contain a short summary, limits, further research... I see you put some of these above in Discussion section.

Response 2- (6)

According to the advice of reviewer, we have revised the conclusion section. (Lines 480-493).

(7) proofread for spaces in the article: F(1, 28) = 75.149

Response 2- (7)

We thank the reviewers for their comments and have now corrected these errors.

(8) CI are reported somehow differently: CI: -0.160 — -0.0137

Response 2- (8)

According to the advice of reviewer, we have revised the format accordingly. (Line 351).

(9) How do you explain that in most significance tests with ANOVA you did not get significant differences?

Response 2- (9)

Thank you for your valuable comment regarding the lack of significant differences in most of the ANOVA tests. The sample size in our study, while appropriate for detecting medium to large effects, may have been insufficient to identify smaller but potentially meaningful differences. These effects, while not statistically significant, may still be biologically relevant and warrant further investigation.

(10) In Bibliography, years are written with bold.

Response 2- (10)

We have revised the format in Bibliography.

Reviewer 3 Report

Comments and Suggestions for Authors

Dear authors. Thank you for allowing opportunity to review this interesting paper. Overall, this is a very informative paper that adds value to the existing research. However, I do have some concerns, mostly around your statistical methods. I offer the below comments and suggestions in good faith.

Author Response

Reviewer 3

Abstract

 (1) Line 11: Your objective could be better defined. I’d suggest expanding on what you mean by procrastination to include active and passive procrastinators. As it stands it is confusing.

Response 3- (1)

Thanks for the reviewer's comments. The description has been changed. (Lines 11-13).

(2) Line 13: Please define the 190 college students (age + SD, sex, etc).

Response 3- (2)

According to the advice of reviewer, this sentence has been revised. (Line 13).

(3) Line 31: Change ‘there are’ to ‘there were’. You are referring to the past therefore the past tense should be used.

Response 3- (3)

Thanks for the reviewer's comments. We have revised this error. (Line 26).

 Introduction

(4) Line 42: To say that the essence of procrastination is a failure in self-regulation, is a big statement. Do you have a citation to back this up? Procrastination can, and does, occur for varying reasons and for varying durations.

Response 3- (4)

According to the advice of reviewer, this sentence has been revised. (Lines 46-47).

(5) Line 48-51: A citation is provided for passive procrastination but not for active procrastination. Why only one reference? Have you defined what active procrastination is or could be without drawing on an exact definition?

Response 3- (5)

Thanks for the reviewer's comments. We have added relevant references.

(6) Line 59: What is meant by ‘switching and updating’?

Response 3- (6)

According to the advice of reviewer, this sentence has been revised. (Line 74).

(7) Line 65: I’d strongly suggest that you define what ‘executive function’ represents. For instance, consider cognitive control for solving problems, managing emotions, etc.

Response 3- (7)

According to the advice of reviewer, we have clarified the definition of 'executive function' in the manuscript. (Lines 65-72).

(8) Line 66: This isn’t the correct definition of physical exercise. See here for a good definition: https://pmc.ncbi.nlm.nih.gov/articles/PMC1424733/

Response 3- (8)

According to the advice of reviewer, we have revised the use of 'physical activity' and 'exercise' in the manuscript." (Line 80).

(9) Line 68: Same here when talking about physical activity. It is important to clarify the differences between exercise and physical activity.

Response 3- (9)

According to the advice of reviewer, we have clarified these expressions. (Line 82).

(10) Line 71: Just the intensity level? What about the duration? The type of activity undertaken?

Response 3- (10)

Relevant literature has shown that the intensity and duration of physical activity can influence procrastination behavior, and we have included this discussion in the manuscript, but the type of activity undertaken. (Lines 81-91).

(11) Line 77: I would suggest that you carefully review the descriptions of physical activity and exercise as to say that ‘that there is a significant negative correlation between physical exercise and procrastination’ is misleading given the terminology.

Response 3- (11)

According to the advice of reviewer, we have revised these expressions. (Line 92).

(12) Line 82: Isn’t this because assessing the role of executive function is difficult to assess?

Response 3- (12)

Assessing the role of executive function indeed presents certain challenges, such as the complexity of executive function itself, the limitations of measurement tools, and the interference of individual differences. These factors may be among the main reasons why its role has not been fully recognized in procrastination research.

(13) Line 92-93: Please clarify the aims of your research. In one of your aims, you mention college students yet this isn’t the case in the other aim. Also, the citations that you have used, good as are, refer to adults and not college students. Consider drawing on this lack of research as it would appear there is a research ‘gap’.

Response 3- (13)

According to the advice of reviewer, we have revised these expressions. (Lines 107-115).

Materials and Methods

(14) Line 100-101: How were students selected? What was the recruitment process?

Response 3- (14)

In this study, we recruited participants by distributing questionnaires to first- and second-year students enrolled in 20 classes taught by 3 physical education instructors. Participation was voluntary, and a total of 455 students were recruited through this process.

Exercise Intervention Program

(15) Line 131: Why was running selected? This appears to be an odd selection. Where any participants classified as ‘runners’? Were baseline activity measures taken?

Response 3- (15)

Given the diverse skill levels of students in other forms of physical activity and their varying academic schedules, running was chosen as the intervention method because it is not constrained by location, time, or group size. Prior to the experiment, baseline activity levels were assessed for all participants, and individuals who reported regular exercise habits were excluded from the study.

(16) Line 133: What was ‘adaptive’ training? A description should be included. Also, who supervised adherence to these sessions?

Response 3- (16)

According to the advice of reviewer, we have added these descriptions (Lines 163-164). In this study, we required all participants in the intervention groups to use a fitness application (Keep App) to record each exercise session and upload it to a WeChat group. The researchers of this experiment supervised and guided the exercise process.

Executive Function Testing

(17) Line 144: Interesting use of the Stroop task. Has this been used to assess procrastination before?

Response 3- (17)

Reference: Carlson SE, Suchy Y, Baron KG, Johnson KT, Williams PG. A daily examination of executive functioning and chronotype in bedtime procrastination. Sleep. 2023 Aug 14;46(8): zsad145. doi: 10.1093/sleep/zsad145. 

(18) This section is interesting and detailed, but I am left to wonder what level of instruction was provided to the participants; was a trial (test, familiarization) session permitted?

Response 3- (18)

Prior to each task assessment, the examiner instructs the test subjects to practice the task until they achieve an accuracy rate of no less than 85%, after which they proceed to the formal testing phase. (Lines 213-215).

Data Analysis

(19) Line 172: Did you measure intensity via HR response or similar? If so you could have used a two-way ANOVA. Or if looking at age-specific data a two-way ANOVA could have been used.

Response 3- (19)

We would like to express our gratitude to the reviewer for their valuable suggestions. In future research, we plan to further investigate the impact of various factors, such as exercise intensity, gender, and cultural differences, on the improvement of executive functions in individuals with different types of procrastination through physical activity. We will rigorously control relevant variables to ensure the rigor and scientific validity of our experiments.

Results

(20) Table 1: The rightmost column is very difficult to interpret. Can this be improved?

In Table 1 and Table 2 there are very large SD, so I’m not sure how much value they are adding as it’s clear that there is much dispersion. Perhaps consider using a Standard Error?

Response 3- (20)

According to the advice of reviewer, we have revised the rightmost column in Table 1. Besides, we sincerely appreciate your valuable feedback on the presentation of data in our study. The relatively large standard deviation indeed reflects a certain degree of dispersion within the data, which is an authentic representation of our research sample. The reason we have chosen to retain the standard deviation is that it provides a more intuitive reflection of the distribution of the data.

Homogeneity Test for Pre-Test Executive Function

(21) Line 206: There are many issues here. Firstly, was homogeneity tested for? T-tests are mentioned throughout yet these were not defined in your statistical methods.

Response 3- (21)

The homogeneity of executive function levels across the different groups before exercise was tested. According to the advice of reviewer, we have added the definition of t-tests in the statistical methods section. (Line 220).

(22) Line 207-210: This reads more like a methods section and not results. I would consider moving this paragraph.

Response 3- (22)

According to the advice of reviewer, we have revised this section.

Impact of Aerobic Exercise Intervention on Executive Function in Students with Different Types of Procrastination

(23) Line 231: This is very confusing I’m afraid. You now mention that a three-way ANOVA was completed, but this was not mentioned at all in your statistical analysis. Please clearly specify what statistical approaches were used. If a three-way ANOVA was used the variables much be defined.

Response 3- (23)

According to the advice of reviewer, we have revised this section (Line 222). The relevant variables for the three-way ANOVA are described in section 2.2 of the manuscript. (Lines 148-150).

(24) Table 3. It’s unusual to include the Sum of Squares and the degrees of freedom in results (and again noting that this wasn’t defined in your statistical approach). What is the justification for including this? It’s not discussed nor highlighted in your results.

There are also lots of formatting issues whereby scientific format (for numbers) hasn’t been set as there are ‘E’ references in your figures (e.g., Updating Accuracy/ Group × Time). Please ensure that this is corrected as results cannot be interpreted as it stands.

Response 3- (24)

We sincerely appreciate your valuable feedback on the presentation of results in our study. Regarding the inclusion of the Sum of Squares and degrees of freedom in the results, we understand that this may not be common practice in some research fields. However, in our study, this information is provided to offer a more comprehensive detail of the statistical analysis, enabling readers to gain a deeper understanding of the ANOVA results.

According to the advice of reviewer, we have defined the Sum of Squares and the degrees of freedom in the statistical approach section (Lines 224-225).

1.517E-005 is equal to 1.517 multiplied by 10 to the -5th power, i.e. 0.00001517, which is displayed as 1.517E-005 because the need to retain 3 decimal places makes the number 0.

Interference Inhibition Task—Stroop Task

(25) Line 237: You mention that “reaction time and accuracy the Stroop Task” was assessed. This should be included in your methods.

Response 3- (25)

The reaction time and accuracy of the Stroop Task included in 2.3.2 section. (Lines 174-176).

(26) Line 243: You’ve mentioned both f and P values. The P value was defined but there was no mention of the f value. I don’t believe you need to include both values.

Response 3- (26)

In the data analysis report, we report both the F-value and its corresponding P-value in order to ensure the transparency and verifiability of the results.

(27) Line 276-278: I would suggest that you carefully revisit your methodology and statistical approach. Pairwise comparisons and confidence intervals are mentioned

Response 3- (27)

Pairwise comparisons and confidence intervals are mentioned here because there was a two-factor interaction in conducting the three-factor repeated measures analysis, so two-by-two comparisons were made, which is specific to three-factor repeated measures ANOVA, and post-hoc comparisons were conducted using the Bonferroni method was mentioned in the statistical analysis section.

(28) Line 286: This sentence is confusing and unclear.

Response 3- (28)

According to the reviewers' comments, we have revised this sentence to make it clearer. (Lines 342-346).

(29) Line 289: You state that ‘when controlling for group …”. What control? Please define this.

Response 3- (29)

According to the reviewers' comments, we have revised this expression. (Lines 322-323, Lines 327-328, Lines 346-347).

(30) Line 315. There’s a reference missing as Chu and Choi is mentioned but no numerical citation is provided.

Response 3- (30)

According to the reviewers' comments, we have added relevant reference.

(31) Line 336: Coping strategies are mentioned here. Were coping strategies examined? This is an interesting addition in that successful coping strategy may influence results. Did you ask participants about coping strategies?

Response 3- (31)

Thank you for your careful review and valuable feedback on our study. We fully agree with your perspective that successful coping strategies may have a significant impact on the research outcomes. In the current study, we primarily focused on the relationship between executive function and procrastination behavior. However, we did not directly conduct a systematic investigation into coping strategies, nor did we specifically inquire about participants' coping strategies in the experimental design. This is a limitation of our research. In future studies, we will consider incorporating systematic measurements of coping strategies.

(32) Line 338: Exercise or physical activity?

Response 3- (32)

Thank you for the reviewer's comment. We have revisited the literature, and it should indeed be "exercise" here.

(33) Line 379: You haven’t specified what limitations exist.

 Response 3- (33)

According to the reviewers' comments, we have added the limitations of this study to the manuscript. (Lines 460-472)

Round 2

Reviewer 1 Report

Comments and Suggestions for Authors

In my opinion, the authors did a good job of taking my review comments into account.

Reviewer 2 Report

Comments and Suggestions for Authors

Congratulations for all changes. They are significant.